# WT and A53T α-Synuclein Systems: Melting Diagram and Its New Interpretation

**DOI:** 10.3390/ijms21113997

**Published:** 2020-06-03

**Authors:** Mónika Bokor, Ágnes Tantos, Péter Tompa, Kyou-Hoon Han, Kálmán Tompa

**Affiliations:** 1Institute for Solid State Physics and Optics, Wigner Research Centre for Physics, 1121 Budapest, Hungary; tompa.kalman@wigner.hu; 2Institute of Enzymology, Research Centre for Natural Sciences, 1117 Budapest, Hungary; tantos.agnes@ttk.hu (Á.T.); peter.tompa@vub.be (P.T.); 3VIB-VUB Center for Structural Biology, Vrije Universiteit Brussel, 1050 Brussels, Belgium; 4BioMedical Translational Research Center, Korea Research Institute of Bioscience and Biotechnology (KRIBB), Rucheng District, Dade District 305 333, Korea; khhan600@kribb.re.kr

**Keywords:** NMR spectroscopy, bond energy, hydration, proteins, aggregation

## Abstract

The potential barriers governing the motions of *α*-synuclein (*α*S) variants’ hydration water, especially energetics of them, is in the focus of the work. The thermodynamical approach yielded essential information about distributions and heights of the potential barriers. The proteins’ structural disorder was measured by ratios of heterogeneous water-binding interfaces. They showed the *α*S monomers, oligomers and amyloids to possess secondary structural elements, although monomers are intrinsically disordered. Despite their disordered nature, monomers have 33% secondary structure, and therefore they are more compact than a random coil. At the lowest potential barriers with mobile hydration water, monomers are already functional, a monolayer of mobile hydration water is surrounding them. Monomers realize all possible hydrogen bonds with the solvent water. *α*S oligomers and amyloids have half of the mobile hydration water amount than monomers because aggregation involves less mobile hydration. The solvent-accessible surface of the oligomers is ordered or homogenous in its interactions with water to 66%. As a contrast, *α*S amyloids are disordered or heterogeneous to 75% of their solvent accessible surface and both wild type and A53T amyloids show identical, low-level hydration. Mobile water molecules in the first hydration shell of amyloids are the weakest bound compared to other forms.

## 1. Introduction

The studied proteins play an essential role in Parkinson’s disease (PD), which is an age-related neurodegenerative disorder diagnosed by tremor at rest, rigidity, and bradykinesia symptoms. It affects more than 5% of the aged population worldwide (above age 85) and is a challenging problem in modern society [1]. Aggregation of the *α*-synuclein (*α*S) protein is thought to be a critical step in the pathogenesis of PD and several other neurodegenerative disorders. Human *α*S is a 140-residue, highly conserved presynaptic protein that is abundant in various regions of the brain [2,3]. Structurally, purified *α*S is a monomeric intrinsically disordered protein (IDP) at neutral pH [4,5]. The phrase IDP refers to proteins without fixed three-dimensional structures under physiological conditions. Due to their specific amino acid compositions, IDPs, in general, have a unique combination of low overall hydrophobicity and large net charge [6]. Though *α*S is an IDP, it shows secondary structure to a limited extent [7,8,9] and is slightly more compact than a random coil [8,10]. Furthermore, high-resolution NMR analysis revealed that it exhibits a region with a preference for helical conformation [10].

αS variants were investigated in the forms of monomers, oligomers, and amyloid fibrils. A question to answer in this work is the phenomenon of aggregation and the role hydration-water plays in this. We wish to explore the structural differences between wild type (WT) and A53T *α*S variants, as well. A further aim of our studies is the investigation of the potential barrier distributions around the different *α*S variants at the actual thermal energy. Wide-line ^1^H-NMR was used as a tool to record melting diagrams (*MD*, amount of mobile hydration vs. normalized fundamental temperature) of the frozen protein solutions [11,12,13,14] to fulfil these aims.

Wide-line ^1^H NMR results are interpreted in a thermodynamic approach as novelty in this work. This approach was introduced in our earlier publications [11,12,15,16,17,18,19]. We have made significant advancements in our understanding since publication [4]. We can get information on the energetics of mobile-hydration-water binding properties of proteins due to the advancements. Thermodynamical properties (e.g., fundamental thermal scale, order parameters, surficial potential distribution of proteins) are used in the description. The novel information obtained in this work includes the distributions of the potential barriers, which control the motion of the mobile hydration water. Reorienting water molecules, which are also bound to a protein molecule, are mentioned as mobile hydration when they reorient fast enough to be seen as mobile by NMR. Through the distribution of the potential barriers, information is gained about the strengths of the protein-water bonds. The distribution of the different types of potential energies (different in magnitude and heterogeneity) shed light on the disordered or ordered global nature of the protein. The amounts of bound mobile water in different states are determined. Quantities of water bound to heterogeneous or homogeneous parts of the solvent-accessible surface (SAS) of the proteins are distinguished and measured.

The potential barriers reflect the protein-water interactions and vary in chemical and topological properties of the interactive SAS of the protein. The measurement and the interpretation process we apply for *α*S monomer variants make also possible to investigate molecular interactions. Oligomerization and amyloid formation are important phenomena in disease occurrence. We investigate the role hydration water plays in these. The degree of hydrophilicity of the protein’s interaction sites determines the quantity of hydration. Hydrogen bonds between protein and water play a dominant role in the formation of the first hydrate shell.

Disease-related mutant forms of *α*S have been identified. Alanine-to-threonine substitution at the 53rd amino acid residue entail inverse preferences of alanine to form helices and of threonine to support *α*-sheet structures, which are crucial for amyloid fibril formation. Accordingly, the A53T variant, which is an early onset mutant, was predicted to be more likely to form β-sheet structure than WT *α*S [20,21]. A53T *α*S forms fibrils the fastest in vitro [22], and WT *α*S is the slowest in fibril formation of all the WT and familial mutant variants (A30P, E46K, A53T). *α*S oligomers, which are thought to be the precursors of fibrils are more toxic than mature fibrils. A53T mutation has been shown to alter the neurotoxicity and aggregation properties of the WT *α*S protein [21]. Dynamics of the proteins play an important role in the process of amyloid fibril formation [22]. The WT and A53T *α*S monomers exhibit almost identical structural properties and conformational behaviour [3,23,24]. The oligomers exist in a range of sizes, with different extents and nature of β-sheet content and exposed hydrophobic surface [25]. They contain 35 ± 5% of β-sheet structure, as opposed to 0% in the soluble *α*S monomers and 65 ± 10% in the *α*S fibrils [26]. We showed earlier that the oligomeric form is ordered, similarly as the globular proteins are [4]. The molecular mechanisms underlying *α*S aggregation remain unknown [21,27].

We compared the details of the hydration of WT and A53T *α*S previously by a combination of wide-line ^1^H NMR, differential scanning calorimetry, and molecular dynamics simulations in publication [4]. We have demonstrated that wide-line ^1^H NMR spectrometry can distinguish point-mutants based on the water-binding heterogeneity of the protein surface [11,15]. It was suggested that a hydration shell of *α*S monomer could be compatible with largely disordered states [4]. In this previous work, it was revealed that the hydration curves of *α*S have features characteristic of IDPs, though the measurements presented in reference [4] were not focused on the structures of the proteins. They presented an interpretation process based on the measurement of the mobility of hydration water. The monomer A53T *α*S mutant has bias for more open structures, with the potential of binding more water molecules. We drew quantitative conclusions about the ratios of the ordered and disordered (more solvent-exposed) surface regions of proteins [15]. The extent of the disordered regions and the energy relations of the protein-water bonds could be determined. Molecular dynamics simulations showed that *α*S behaviour measured by wide line ^1^H NMR corresponds to the first hydration layer of the structural ensemble of the protein. It was found that the first hydration shell around globular proteins is more compact than around disordered proteins [4]. Moreover, *α*S amyloids have some residual structural disorder even in the highly ordered aggregated/amyloid state. The differences between WT and A53T *α*S monomers disappear in the amyloid state and they show identical *MD*s, suggesting the same surface topology, irrespective of the initial monomeric state.

As a novelty, we use the interpretation of the *MD*s to inform about the thermodynamic nature of the SAS of the protein in this work. We present here an energetic description of the potential barriers governing motions of water molecules in the hydration shell of *α*S [11,15]. We introduced normalized fundamental temperature and an energy scale was set up in the form of normalized fundamental temperature being scaled with the molar enthalpy of fusion for water. The normalization is made by the melting point of bulk water. The dynamic order parameter of the *MD*, i.e., the heterogeneity ratio characterizes protein-water-bond energy distribution. The usage of fundamental temperature helps to describe *MD* in the form of power series. The derivative form of the latter defines the number of water molecules that begin to move at a given potential barrier.

The amount of mobile hydration water is calculated from the amplitude of the wide line ^1^H NMR signal. In this work, only a noticeably short description is provided, for more details of the measuring method, see reference [15] and *Supplemental Information*. We could set up energy distributions of the potential barriers affecting the motion of hydration water, for better understanding and deeper insight.

## 2. Results

The first hydration layer on the protein surface is of particular importance for biological activity [13]. A protein with its first hydration layer forms the biologically active entity [14]. Hydration water moves on the picosecond time scale while NMR-processes are slower by an order of magnitude. The phenomenon of motional narrowing in NMR, therefore, is a good indicator of mobile water.

Proteins do not interact directly with the bulk water phase. Their interactions are realized with a few bound [28] or non-freezing water layers. This fact rationalizes studying the properties of bound water. Bound or non-freezing water is referred to here as mobile hydration water. Hydration water must be regarded not as a rigid shell of water molecules around the protein molecules, but as a fluctuating cloud of water molecules that interacts more or less strongly, even unfavourably with the protein surface. The protein-water interaction energies are considered to equal with the relevant potential barrier values for the mobile hydration water. The extent of the exposed hydrophobic surface determines how strongly water is bound in the hydration shell of a protein [29]. It is inversely proportional to bond strength. The motion of the hydration water molecules, which are the most loosely bound to the proteins is hindered by potential barriers quite different in magnitude for the different forms of *α*S.

### 2.1. α-Synuclein Monomers

The disordered nature of IDPs originates from a large content of hydrophilic amino acid residues compared to hydrophobic ones, preventing the hydrophobic collapse [30]. The WT and A53T variants of *α*S monomers dissolved in pure water show *MD*s characteristic of IDPs as follows (Figure 1a) [4,11,15,16,18,31]. (i) Mobile hydration water molecules appear around these proteins at a relatively high potential barrier value as the lowest potential barrier value where mobile hydration water exist (*E*_a,0_, Table 1). The actual *E*_a,0_ counts as especially high for *α*S monomers when compared to oligomer and amyloid *α*S forms or to globular proteins [11,15]. (ii) Moreover, the *MD*s of IDPs have a steeply increasing trend with rising potential barrier values from E_a,1_ to *E*_a_ = 6.01 kJ mol^−1^ (0 °C; Figure 1a) [4,11,15,16,18,31].

The *MD*s of WT and A53T *α*S monomers show rapidly changing trends as a function of potential barriers governing the motion of hydration water, except for a short constant region at the start of mobility (Figure 1a,d). The amount of mobile water molecules, *h* = 0.44(8) is enough to have functional proteins with approximate monolayer coverage at the smallest potential barriers below *E*_a,1_, where a constant *h* region can be found (for definition of hydration see Section 4.3*. Interpretation methods of the melting diagram*). The mobile water concentration given as hydration provides a macroscopic measure. This level of mobile hydration water is equal to 3.5(6)·10^2^ H_2_O/protein on the microscopic scale. While *h* given in g water/g protein units, measures the amount of mobile water on a macroscopic scale. Molecular dynamics simulations show that the measured behaviour corresponds to the first hydration layer of the structural ensemble of the protein [4]. Earlier NMR measurements and molecular dynamics simulations for individual amino acid’s hydration yield 4(1)·10^2^ water molecules per amino acid sequence of the whole protein [30,32,33]. The comparison between our results and the predicted values (red line, Figure 1a) suggests that the solvent accessible amino acid residues are fully hydrated with mobile water molecules in the monomers as soon as water molecules are no longer rigid in the ice phase. The amount of homogeneously bound hydration water, *n*(*T*_fn0_) = *n*(*E*_a,0_) is as high as if every possible hydrogen bond site were occupied by a water molecule. This refers to an open structure.

Water in interaction with A53T *α*S is set in motion at a lower potential barrier compared to the WT variant (Figure 1a,d). This fact indicates that A53T *α*S interacts with water more weakly, and that its protein-water bonds are looser. The constant-value *h* regions at low potential barriers (from *E*_a,0_ to *E*_a,1_, Table 1) indicate the presence of secondary structures even though these proteins are IDPs. To support this statement, see the *MD*s of the two globular proteins, UBQ and BSA dissolved in water and the *MD*s of IDPs [11,15]. The globular proteins have *MD*s consisting of almost only temperature-independent hydration while the IDPs have constantly increasing *MD*s with no or very narrow temperature-independent hydration. The occurrence of secondary structures in IDPs is not unexpected [34]. These regions appear as spikes and subsequent zero values in differential forms of melting diagrams (*DMD*s). Consequently, only one type of potential barrier or protein-water interaction is responsible for the motion of water molecules in this energy region. The narrow energy spectrum of this feature is not consistent with a van der Waals interaction, which would have a wider range of interaction energies, and suggests instead the formation of hydrogen bonds.

The plateau region (constant value of *h*, *E*_a,0_ ≤ *E*_a_ ≤ *E*_a,1_) is wider for A53T *α*S but the *E*_a,1_ values are the same for both variants (Table 1). This means that A53T *α*S has a wider spread of non-occupied protein-water interaction energy levels and the potential barrier gap is wider. The explanation for this can be that the A53T mutation results in a local preference for extended backbone conformations at a short contiguous region at around residue 53 [35].

WT and A53T *α*S monomer proteins are intrinsically disordered and that is why their *MD*s show rapidly changing thermal trends above *E*_a,1_ (Figure 1a,d). Solvent-exposed amino acid residues result in a SAS of the protein highly heterogenic by energy. Slope of *MD* for *α*S monomers in the linear section (*E*_a,1_ < *E*_a_ < *E*_a,2_) is more than three times steeper than in the case of *α*S amyloid (parameter *B*, Table 1). The appearance of heterogeneous distribution of potential barriers at *E*_a,1_ is connected with general change in the motional state of the hydration water. In general for both globular proteins and IDPs, dynamic contributions to the quasi-elastic neutron scattering spectra [14] all change their thermal trend around *E*_a,1_. At the higher hydration levels, not only hydrogen bonds are responsible for protein-water interactions but van der Waals forces are also acting. Above *E*_a,2_, the *MD*s for both *α*S monomer variants showed a quadratic trend(Figure 1a)

The *α*S monomers have the highest ratio of exposed hydrophilic groups with more than five times more moving water in their first hydration shell than the *α*S amyloids (A = *n*(*T*_fno_) = *n*(*T*_fn1_) or *h*(*E*_a,o_) = *h*(*E*_a,1_), Table 1.).

The *DMD* (Figure 1d–f) provides a quantitative measure of the energetic heterogeneity of the potential barriers affecting the mobility of hydration water molecules. The *α*S monomer-water interaction energy is most diversified. This can be understood as resulting from the open structure of the monomers, with many different types of hydrophilic groups in their water accessible surface. The difference between the *MD* of the WT and A53T αS monomers is very minute (Figure 1a), though the A53T mutant is steadily more heterogeneous (*C*, Table 1).

### 2.2. α-Synuclein Oligomers

Features common to the different types of *α*S oligomers include β-sheet structures, high content of solvent-exposed hydrophobic regions and globular or tubular morphology [25,36]. The general surface of the *α*S oligomers is essentially the same irrespective of their size, which varies. This uniformity is assumed because there can be seen but only one type of oligomers below *E*_a,3_, according to the *MD*s.

The oligomer forms of WT and A53T *α*S both show *MD*s similar to those of globular proteins [11,15]. This manifests itself in attributes as (i) the E_a,0_ is much lower than in the case of the IDP monomers; (ii) the wide temperature-independent region in their MDs; and (iii) steep rise in the hydration level shows up at high potential barriers near to *E*_a_ = 6.01 kJ mol^−1^.

The *MD*s of the oligomeric *α*S forms (Figure 1b,e) are dominated by a wide constant hydration region. This is a general characteristic of ordered (folded) structure present in the *α*S oligomers. The very wide constant *h* region can be caused by the greater extent of solvent exposed hydrophobic surface [25]. There is only one constant *MD* region with one single *h* value. This fact means that there are no water molecules, the motion of which is controlled by potential barriers with energies *E*_a,0_ < *E*_a_ < *E*_a,1_ = *E*_a,3_. The constant *h* region in the *MD* of WT *α*S oligomers is wider than that in the *MD* of A53T *α*S oligomers. Both are far wider than that of the IDP *α*S monomer forms. One single potential barrier (or a very narrow distribution of it) is characteristic for the motion of the *h*(*E*_a,0_) amount of mobile water, meaning a homogeneous energy distribution of potential barriers for them. These water molecules are homogeneous in their bonding properties since no others start to move in an overly broad potential energy range.

The *α*S oligomers have a significant surface ratio of hydrophilic nature, which manifests itself in the wide heterogeneous *MD* section just below *E*_a_ = 6.01 kJ mol^−1^ (the growth can be described with a cubic term in the *MD*s from *E*_a,3_ on). Our results may reflect the variety in oligomer *α*S size as the potential barrier range with an intensively growing amount of mobile water is relatively wide compared to globular proteins [11,15]. That is, a very heterogeneous distribution of the actual potential barriers can be found for the water molecules starting to move in this energy range. Not only the amount of mobile hydration water grows rapidly, while it approaches the melting point of bulk water, but the intensity of the growth also increases.

The *α*S oligomers’ ordered nature is further confirmed by the very wide constant section of the *MD*s. The mobility in their hydration shell is more limited than for the monomers. *α*S oligomers show homogeneous protein-water interactions till *E*_a,1_ = *E*_a,3_ and their hydration is low at this potential barrier level, *h(E*_a,3_) = 0.32(2) (on the average). *E*_a,3_ is the beginning of the *MD*’s cubic-increase section. In a narrow, 0.363(5) kJ mol^−1^ wide energy interval, *α*S oligomers reach a relatively high hydration level by *E*_a_ = 6.01 kJ mol^−1^, *h*(*E*_a_ = 6.01 kJ mol^−1^) = 2.1(1). This behaviour can also be a sign of their variety in size. The growth of hydration is Δ*h* = 1.760(4) for *α*S oligomers in the region of heterogeneous elevation, at *E*_a,3_ < *E*_a_ < 6.01 kJ mol^−1^, that equals the size of the heterogeneously bound water fraction, *n*_he_ (Table 2). *HeM* values of oligomers are the highest indicating an intensive growth of bond heterogeneity in this region, starting from a low level. Hydrations of *α*S oligomers at *E*_a_ = 6.01 kJ mol^−1^ are much lower than that of *α*S monomers here and are a little greater than that of *α*S amyloids.

### 2.3. α-Synuclein Amyloids

Fibrillary conformation of *α*S is largely composed of parallel β-sheet structures [25,37]. The hydrate shell of the *α*S amyloids can be mobilized the most easily, which feature is related to the weak bonds holding together the parallel β-sheets. Water bound to *α*S oligomer molecules requires higher energy to move and the hydration water of the *α*S monomers can be moved at the highest energy cost. The mutations associated with familial PD (e.g., A53T) increase the propensity of *α*S monomer to form insoluble aggregates and produce morphologically distinct aggregate species according to literature data [38,39,40,41,42]. The amyloid state shares distinctive structural features that are largely independent of the protein sequence [43,44]. We could see identical *MD*s of the WT and A53T amyloid variants (Figure 1c,f) in accordance with this feature and therefore, they are discussed unified.

The quantity of mobile hydration water at the start of motion is by far the smallest in amyloids compared to other *α*S states so it can be assumed that they have the greatest extent of exposed hydrophobicity. This statement is supported by the following reasons. Hydrophobic protein surfaces do not attract water and therefore a regular net of hydrogen bonds does not order water near them. This chaotropic water structure is not extended to large distances from the surface and the involved water molecules are not really bound to the protein. The amyloids have *h*(*E*_a,0_) = 0.077(6), a value that is approximately six times smaller than for *α*S monomers and more than four times smaller than for *α*S oligomers (Table 1, Figure 1a–c). The exceptionally low *E*_a,0_ is a characteristic feature of amyloids, even globular proteins show higher potential barrier values as the lowest where mobile hydration water exists [11,15]. This implies that significant solvent space exists within the fibrils, which is attributed to the *α*S molecules within the fibrils having a distribution of conformations. The water inside the structure cannot be seen as mobile by wide line ^1^H NMR due to limited mobility. This fact can be responsible for the small amounts of mobile hydration water around the *α*S amyloids.

The low *E*_a,0_ and the subsequent zero value region in the *DMD* refer to hydrophobic patches of the SAS of the *α*S amyloid fibrils as such surface parts does not establish H-bonded protein-water interactions. The constant mobile hydration region in the *MD* is half as wide as in the *α*S oligomers, but it is still twice as wide as in the *α*S monomers. The hydrophobic patches relate to secondary structures. Mature fibrils contain several protofilaments with a cross β-structure, in which individual β-strands run perpendicular to the fibril axis [45,46,47,48,49]. Only specific segments of the chain are incorporated into the cross-β structure, with the remaining of the chain being external to the core elements of the structure. The central portion of *α*S corresponding to amino acid residues 31-109 (half the size of the holoprotein) constitutes the core of the *α*S amyloid filaments [26].

The abundant solvent-exposed side chains [22] are most possibly responsible for the mainly heterogeneous potential barrier distribution affecting the motion of the mobile hydration water at *E*_a,1_ < *E*_a_ < 6.01 kJ mol^−1^ (Figure 1c,f), and therefore they are responsible for the heterogeneous protein-water interactions. Further properties that increase heterogeneity are the large amplitudes of the side chain motions (larger in the fibril state than in the monomeric state, the amyloid state is entropically favourable [22]) and heterogeneity is also increased by the fibrils having a distribution of conformations. The SAS of the *α*S amyloid form can be considered as heterogenic in its protein-water interactions to 75(3)% or conversely, it bonds water homogeneously to a measure of 25(3)% regarding the nature of its SAS (*HeR* value, Table 2).

In an assembled *α*S amyloid, long N-terminal and C-terminal segments remain unprotected (residues 1−~38 and 102–140), although the N-terminal segment shows some heterogeneity. A continuous middle segment (residues ~39–101) is strongly protected by a systematically hydrogen bonded cross-β structure [45]. The termini of the *α*S monomers are much more charged (net charge is −10 with 26 pieces of charged residues) than the middle (net charge is +3 with 13 pieces of charged residues). The number of H_2_O molecules bound to the protein [30,32,33] is 2.4(6)·10^2^ for the termini and it is 1.5(3)·10^2^ for the middle section, as summed for the individual amino acids. The same number for the whole *α*S is 4(1)·10^2^. Nominally, the terminal sequence binds 55% of the water molecules compared to the whole protein. Since the middle segment forms the β-sheet structure and therefore it is not involved so much in hydration, these hydration ratios can explain that the mobile hydration of the *α*S amyloid fibrils is approximately half of that of the *α*S monomers.

### 2.4. Dynamic Parameters and Comparisons

Both WT and A53T *α*S monomers reach their highest hydration level at the melting point of bulk water, *h*(*E*_a_ = 6.01 kJ mol^−1^, monomer) = 3.31(7) on the average, exceeding the values observed for oligomers and amyloids, and indicating a mostly solvent-exposed polypeptide chain. Intramolecular interactions in the WT αS monomer structure inhibit fibril formation. These interactions are greatly destabilized by the A53T mutation [50], which shifts the conformational ensemble of WT αS monomer to a more open state [51]. The observed heterogeneity ratio (*HeR*, Table 2) is not 1 as expected if αS monomers were totally disordered but it scores 33(4)% homogeneous potential barrier distribution for WT and A53T *α*S. This fact is in accordance with the structural ensemble of *α*S being more compact than it is expected for a random coil state [52,53]. The ratios of heterogeneously bound mobile water fractions to the sum of both heterogeneously and homogeneously bound water, *HeR*_n_ values are equal within experimental error for all of the *α*S variants, except *α*S amyloids (Table 2). *α*S monomers have, nevertheless, little higher *HeR*_n_ values than *α*S oligomers and *α*S amyloids are characterized by far the highest ratio of heterogeneously bound mobile hydration water. *HeR*_n_ = 0.87(1) is calculated as average for the monomer and the oligomer variants, indicating high heterogeneity in the nature of the bonds formed by these *α*S variants. By this ratio, *α*S amyloids show the most heterogeneously bound water molecules in this region near to the melting point of bulk water. The average amount of heterogeneously bound water for the monomers, *h*_he_^monomer^ = 2.8(2) is much higher than for oligomers (the averages are for oligomers *h*_he_^oligomer^ = 1.760(4)) and for amyloids with *h*_he_^amyloid^ = 1.68(9)). This relation of *h*_he_ values reflects that *α*S monomers are IDPs while the *α*S oligomers and *α*S amyloids are significantly more structured. *h*_he_ serves as nominator in the expression of *HeR*_n_. The measure of heterogeneity, *HeM* characterizes the extent of heterogeneity of the protein-water interactions close to *E*_a_ = 6.01 kJ mol^−1^. WT and A53T *α*S monomers show *HeM* values an order of magnitude smaller than *α*S oligomers and *α*S amyloids. The hydration of WT and A53T *α*S monomers is already heterogenic even at *E*_a,2_ and it does not elevate intensively when approaching *E*_a_ = 6.01 kJ mol^−1^.

The small *HeR* values of the WT and A53T *α*S oligomers mean that they are remarkably more ordered than the WT and A53T *α*S monomers (Table 2). *HeR* of the oligomers is well below 0.5 and *HeR*s of the monomers are well above 0.5. WT *α*S oligomers have higher *HeM* values than A53T *α*S oligomers and *α*S amyloids, which means that their bonding network is more heterogeneous close to the melting point of bulk water. The latter two have remarkably similar *HeM* values.

*HeM* values for *α*S amyloids are of similar value to that for oligomers and are much higher than monomers’ *HeM*. Bonding heterogeneity and *h*(*E*_a_ = 6.01 kJ mol^−1^) levels are quite similar for *α*S amyloid and oligomer variants. The main difference between these two aggregated species is that *α*S oligomers are ordered in contrast with disordered *α*S amyloids. The hydration value at the melting point of bulk water, *h*(*E*_a_ = 6.01 kJ mol^−1^) for the *α*S amyloids is comparable to that of the *α*S oligomeric forms, but it is significantly lower than in the case of *α*S monomers (Table 1). This relation of the relative levels means that not only the *α*S oligomers, but also the *α*S amyloids have a compact structure compared to αS monomers.

## 3. Discussion

*α*S variants are the studied proteins, namely WT *α*S and its A53T point mutant, in monomer, oligomer, and amyloid degrees of polymerization. The bases of our experimental results are the temperature dependences of the number of mobile hydration ^1^H nuclei which generate the narrow-lines of two-component wide-line ^1^H NMR spectra (Figure A1). The resulting melting diagrams are interpreted from a thermodynamic point of view [11,15,18]. The two-component spectra (a narrow and a wide line) originate from the different molecular mobility of water molecules bound to the protein also, not only to other water molecules. The derivative of the melting diagram provides energy distributions of potential barriers hindering the motion of water molecules bound to protein molecules also in aqueous solutions of distilled water.

We demonstrated by direct experimental results that αS monomers are intrinsically disordered proteins but 32(3)% of their SAS is determined by secondary structures. That is, we measured the monomers to be more compact than it is expected for random coils. At the lowest potential barriers, the *α*S monomers have approximately monolayer coverage with mobile hydration water and they are therefore functional. Their SAS is highly heterogenic regarding protein-water interactions. The *α*S monomers have the highest ratio of solvent-exposed hydrophilic groups compared to the *α*S oligomer or amyloid forms. We compared the initial level of *α*S monomer hydration with amino acid hydration data available in the literature, and this comparison showed that every possible hydrogen bond is realized, confirming open structure again. This open nature involves that considering interaction energies, *α*S monomers interact with water most diversified.

One single potential barrier is responsible for the motion of the initial amount of mobile hydration water molecules around *α*S oligomers, which are therefore homogeneous in their bonding properties. *α*S oligomer forms have a uniform general surface; there is only one type of *α*S oligomers below the cubic melting-diagram section. The hydration of the *α*S oligomers is only slightly lower than that of the *α*S monomers in the constant mobile hydration region. The wide, constant melting-diagram region is characteristic of homogeneous protein SAS, homogeneous protein-water interactions. Such protein nature is typical of globular structure [11,15]. The globular structure is most possibly result of the presence of β-sheets. Averagely 66(2)% of the *α*S oligomer variants’ SAS is assumed to be ordered i.e., it has homogeneous protein-water bonds, which involves mostly folded *α*S oligomer states. The mobility of water in the *α*S oligomeric hydration shell is more limited than for *α*S monomers or amyloids. Close to the melting point of bulk water, the melting diagrams of *α*S oligomers reflect their variety in size as a very heterogenic potential barrier distribution in a wide potential-barrier energy range, compared to melting diagrams of globular proteins.

Water molecules in hydration shell of *α*S amyloids can be mobilized the most easily compared to the case of *α*S monomers or oligomers. The hydration of the *α*S monomers is substantially higher in the constant mobile hydration region than in the case of the *α*S amyloids. The identical melting diagrams of the WT and A53T αS amyloid variants show that their structural features are independent of their sequence. The structure of the αS fibril exhibits many stabilizing features of the amyloid fold, including steric zippers involving the hydrophobic side chains and hydrophobic packing [54]. Wide-line ^1^H NMR cannot see the water inside the solvent space within the *α*S fibrils as mobile. We suggest that this can be the cause of the low level of mobile hydration for *α*S amyloids. The potential barrier distribution affecting the motion of mobile hydration water is mainly heterogeneous in the *α*S amyloids, due to the abundant solvent-exposed side chains, the motion of which have bigger amplitudes than in the *α*S monomers [22,55]. Large solvent-filled space within the fibrils is required to this. αS fibrils are disordered except for the core structure [22,55]. In agreement with this fact, we found that the amyloid form of αS can be considered as disordered to 75(3)% regarding the nature of its SAS.

The initial amount of mobile water approximately equals in *α*S monomers and *α*S oligomers, while the *α*S amyloids have several times less such hydration moving at the lowest potential barriers. We could determine that approximately half of the mobile hydration water quantity, measured around *α*S monomers, is lost above the constant mobile hydration section during the formation of *α*S oligomers or *α*S amyloids. The comparison of the melting diagrams of the three αS forms shows this. These water molecules vanishing from the mobile hydration shell due to the aggregation of the monomers are connected to hydrophilic parts of the solvent accessible protein surface. The aggregation buries a significant part of the hydrophilic amino acid residues and these become inaccessible to the solvent water. The buried amino acid residues form most possibly the structural fragments holding the *α*S oligomers and the *α*S amyloids together. These amino acid residues are presumably parts of the β-sheets.

## 4. Materials and Methods

### 4.1. Proteins

Expression and purification of recombinant human wild-type and A53T mutant *α*-synuclein in a pRK-172-based expression system was performed as described in [49]. In sample preparation, the mass of lyophilized protein (without any further refinement) was measured and an appropriate amount of water was added to obtain the requested nominal concentrations. Ultra-pure double-distilled water was used as a solvent in case of monomers, which were measured directly after removing oligomers formed during purification by filtering the solution through a 100 kDa membrane. Amyloid formation was induced by dissolving 50 mg/mL protein in the following buffer: 10 mM Tris-HCl, 50 mM NaCl, 0.02% NaN3, 100 μM CuCl_2_. samples were incubated at 37 ^°^C with 800 rpm shaking and the amyloid fibril growth was followed by ThT fluorescence, with 5 μM ThT (final concentration). Oligomer samples were taken in the lag phase of amyloid growth, after 24 h of the start point. Formed amyloids were separated after 60 h of growth. The presence of fibrils was certified by electron microscopy (Jeol JEM-1011) from samples incubated in the same conditions, but without ThT. Oligomer and amyloid samples were dialysed into ultra-pure double-distilled water before measuring. Final concentrations at the time of NMR measurements were as follows: monomers and amyloids: 25 mg/mL, oligomers: 50 mg/mL. Measurements were carried out on three identical samples prepared independently. All results where a mobile fraction of water, *n*(*T*_fn_) is involved, are calculated for 50 mg/mL protein concentration.

### 4.2. Wide-Line NMR Measurements

^1^H-NMR measurements and data acquisition were accomplished by a Bruker AVANCE III NMR pulse spectrometer at 82.6 MHz frequency, with a stability better than ±10^−6^. The inhomogeneity of the magnetic field was 2 ppm. The pulse width was ~10 µs. The data points in the figures are based on spectra recorded by averaging the signals to reach a signal/noise ratio of 50. The extrapolation to zero time was done by fitting a stretched exponential. The principle and details of wide-line ^1^H NMR spectroscopy are given in refs. [16,31] and in Appendix A.

Free induction decay signals (FIDs) were measured between –70 °C and +25 °C, following thermal equilibrium. The temperature was controlled by an open-cycle Janis cryostat with stability better than ±1 K. The phases of ice protons, protein protons, and mobile (water) protons are separated in the FID signal by large differences in their spin-spin relaxation rate. The portion of mobile proton (water) fraction is directly measured by the FID signal and Carr-Purcell-Meiboom-Gill (CPMG)-echo trains (in the temperature range where echoes could be detected at all) [16,31].

Melting is understood in the frozen protein solutions here, as the appearance of a motionally narrowed component in the ^1^H NMR-spectrum, attributed to mobile water molecules [11]. The melting diagram or curve gives the amount of meltwater–in this sense–as a function of temperature. The amount of meltwater, *n* is measured by the FID amplitude directly as a fraction of the total water content of the protein solution. It is converted into hydration, *h* as gram water per gram protein (Figure 1a–c). Hydration is an intensive quantity, independent of the size or concentration of the protein.

### 4.3. Interpretation Methods of the Melting Diagram

As temperature measurement, fundamental temperature is used, *T*_f_ = *k*_B_*T* or *T*_f_ = *RT*, where *T* is absolute temperature. A normalized version of the fundamental temperature scale was introduced, normalized to the melting point of bulk ice, *T*_fn_ = *T/*273.15 K. The start of a molecular motion can be described in solids by the necessary thermal input as *V*_0_ = constant·*T* [16]. This idea was applied to frozen aqueous protein solutions and was corrected dimensionally [11,15]. The *T*_fn_ scale is converted into an energy scale, scaled with water’s heat of fusion, 6.01 kJ mol^−1^ [56]. The energy scale is given by the relation *E*_a_ = *cRT* on a molecular level, where *c* is a constant. No modelling assumptions are used in the interpretation. Molecular motion is considered to be rotation and not vibration.

Proteins can be characterized and categorized by the heterogeneity of the energy distribution of water binding, described by the heterogeneity ratio
*HeR* = (1 − *T*_fn1_)/(1 − *T*_fno_).(1)

*T*_fno_ and *T*_fn1_ are start and endpoints of the temperature-independent section of the *MD*s. *HeR* provides the ratio of the heterogeneous binding interface (Equation1), and it is a measure of the structural disorder of proteins. Heterogeneous biding means here heterogeneity in bond strengths and bonding partners, bonding type–different types of intermolecular bonds. The *MD*s can be described analytically by a power series as
*n* = *A* + *B*(*T*_fn_ − *T*_fn1_) + *C*(*T*_fn_ − *T*_fn2_)^2^ + *D*(*T*_fn_ − *T*_fn3_)^3^ + …(2)

The derivative form gives the change of *n* and *DMD* (a derivative form of *MD*) is then
∂*n*/∂*T*_fn_ = *B* + 2*C*(*T*_fn_ − *T*_fn2_) + 3*D*(*T*_fn_ − *T*_fn3_)^2^ + …,(3)
and highlights the peculiarities of the hydration melting process.

The ratio of heterogeneous protein-water bonds, *HeR*_n_ [15] is calculated with *n*_ho_, the number of water molecules in the first hydration layer and with the total number of water molecules in the whole of the heterogeneous region, *n*_he_ as
*HeR*_n_ = n_he_ ⁄(n_ho_ + n_he_).(4)

*n*_ho_ is equal with *A*, the number of mobile hydration water molecules at the constant section of the *MD* bound homogeneously. *n*_he_ is number of the mobile hydration water molecules bound heterogeneously, these water molecules are those that become mobile at potential barrier values higher than in the constant *MD*-section. The sum (*n*_ho_+*n*_he_) is equal to the number of the mobile hydration water molecules at 6.01 kJ mol^−1^.

Hydration, *h* =(g water)/(g protein) concentration is used instead of *n* in the text. In order to calculate *h* from *n* (measured by NMR), the following formula can be used *h* = *n*·*m*_water_/*m*_protein_, where *m*_water_ and *m*_protein_ are the masses of water and protein, respectively, in a given volume of the actual solution. 

The measure of heterogeneity, *HeM* describes the heterogeneity of the protein-water bonds near *E*_a_ = 6.01 kJ mol^−1^ and gives fingerprint of each protein. The formula [11,17] is related to the tangent function and to *DMD*,
*HeM* = (*B* + 2*C* + 3*D*)⁄(1 − *T*_fn1_).(5)

For more details on the evaluation and interpretation of ^1^H NMR data, see [11,15] and the *Supporting Information* of [18].

We determined that the mobile hydration water fraction of the wild type and the A53T mutant *α*-synuclein monomers is high enough at the lowest potential barriers to realize every possible hydrogen bond with water.

## Figures and Tables

**Figure 1 ijms-21-03997-f001:**
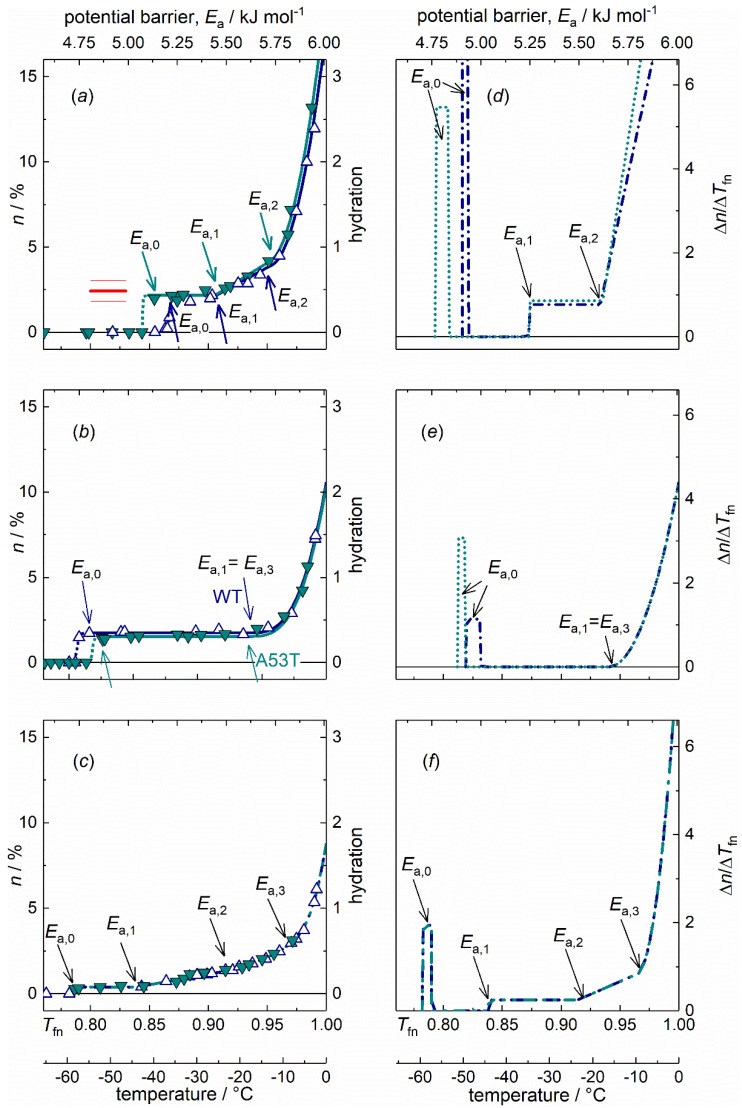
*α*-synuclein **(***α*S) monomers, oligomer and amyloids, wild type (dark blue open up triangles, dark blue dot-dashed lines) and A53T mutant (cyan solid down triangles, dotted cyan lines) variants dissolved in pure water: (**a**) *α*S monomers, (**b**) *α*S oligomers, (**c**) *α*S amyloids, melting diagrams; (**d**) *α*S monomers, (**e**) *α*S oligomers, (**f**) *α*S amyloids, derivatives of the melting diagrams, i.e., potential barrier distributions related to moving hydration water. There are no reliable measured data in the range −1–0 °C (0.995–1.00 *T*_fn_). Data are given for 50 mg ml^−1^ protein concentration. The red line on panel (**a**) is at the predicted hydration value *h* = 4(1)·10^2^.

**Table 1 ijms-21-03997-t001:** Parameter values for the polynomial relation (Equation (2) describing mobile water fraction, *n*. (See Appendix A.) The error in the last digit is given in parentheses.

Polymerization	Monomer		Oligomer		Amyloid
*α*S variant	WT ^1^	A53T	WT	A53T	WT, A53T
*A* = *n*(*T*_fno_) ^2^ = *n*(*T*_fn1_)	0.022(4)	0.022(4)	0.0174(6)	0.0153(4)	0.0039(3)
*h*(*E*_a,o_) ^3^ = *h*(*E*_a,1_)	0.44(8)	0.44(8)	0.35(1)	0.306(8)	0.077(6)
*B*	0.38(5)	0.43(3)	0	0	0.12(1)
*C*	45(7)	0.6(1)·10^2^	0	0	2.9(4)
*D*	0	0	4(1)·10^2^	4(1)·10^2^	10(2)·10^2^
*T* _fno_	0.8662(9)	0.854(5)	0.816(4)	0.8257(2)	0.784(2)
*E*_a,0_/kJ mol^−1^	5.206(6)	5.13(3)	4.90(2)	4.963(1)	4.71(1)
*t*_0_/°C ^4^	−36.5(2)	−40(1)	−50(1)	−47.60(5)	−58.9(6)
*T* _fn1_	0.908(4)	0.906(2)	0.941(5)	0.940(5)	0.838(5)
*E*_a,1_/kJ mol^−1^	5.46(2)	5.44(1)	5.65(3)	5.65(3)	5.04(3)
*t*_1_/°C	−25.2(1)	−25.7(5)	−16(1)	−16(1)	−44(1)
*T* _fn2_	0.951(3)	0.953(4)	—	—	0.914(3)
*E*_a,2_/kJ mol^−1^	5.72(1)	5.73(2)			5.49(2)
*t*_2_/°C	−13.3(8)	−13(1)			−23.5(9)
*T* _fn3_	—	—	0.940(5)	0.940(5)	0.965(2)
*E*_a,3_/kJ mol^−1^			5.65(3)	5.65(3)	5.80(1)
*t*_3_/°C			−16(1)	−16 (1)	−9.6(7)
*n*(*T*_fn2_)	0.039(3)	0.042(8)	—	—	0.013(2)
*h*(*E*_a,2_)	0.77(6)	0.77(3)			0.26(4)
*n*(*T*_fn3_)	—	—	0.0174(6)	0.0153(4)	0.027(1)
*h(E*_a,3_)			0.35(1)	0.306(8)	0.54(3)
*n*(*T*_fn_ = 1)	0.164(5)	0.20(6)	0.105(5)	0.103(5)	0.088(4)
*h*(*E*_a_ = 6.01 kJ mol^−1^)	3.3(1)	4(1)	2.1(1)	2.1(1)	1.75(9)

^1^ Wild type. ^2^ Amount of mobile water fraction at T_fno_. ^3^ Hydration at E_a,0_. ^4^ Temperature in units of degrees Celsius.

**Table 2 ijms-21-03997-t002:** Homogeneously or heterogeneously bound amounts of water, and dynamic parameters. For the detailed definitions of the parameters, see *Supplemental Information* of [18]. The error in the last digit is given in parentheses.

Protein	Variant	*n* _ho_ ^2^	*n* _he_ ^4^	*HeR*	*HeR* _n_	*HeM*
		*h* _ho_ ^3^	*h* _he_ ^5^			
monomer	WT ^1^	0.22(4)	0.142(9)	0.70(4)	0.87(3)	9.8(5)·10^2^
		0.44(8)	2.8(2)			
	A53T	0.22(4)	0.18(6)	0.65(4)	0.89(3)	1.30(6)·10^3^
		0.44(8)	4(1)			
oligomer	WT	0.0174(6)	0.08790(7)	0.32(2)	0.83(4)	2.1(1)·10^4^
		0.35(1)	1.76(1)			
	A53T	0.0153(4)	0.09(1)	0.35(2)	0.85(4)	2.0(1)·10^4^
		0.306(8)	1.8(1)			
amyloid	WT, A53T	0.0039(3)	0.084(5)	0.75(3)	0.96(5)	1.82(9)·10^4^
		0.077(6)	1.68(9)			

^1^ Wild type. ^2^ Mobile water fraction bounded homogeneously. ^3^ Mobile hydration bounded homogeneously. ^4^ Mobile water fraction bounded heterogeneously. ^5^ Mobile hydration bounded heterogeneously.

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
