# Peer review of "WT and A53T α-Synuclein Systems: Melting Diagram and Its New Interpretation"

_ijms, 2020, doi:10.3390/ijms21113997_

Round 1
Reviewer 1 Report
Bokor et al. determined the melting diagram of a-synuclein in its monomer, oligomer and amyloid states. They interpreted the parameters derived from the melting diagram as an indicator of structural order of the protein. However, physical meaning of these parameters were not well described and cross validated with those derived other approaches. In general, this manuscript is difficult to follow, and should be substantially revised.
- L48, is there any other group studied the wide-line 1H NMR?
- L78, alpha-sheet structures?
- L81-82, did the authors mean that among all the WT and mutants, A53T mutant is the fastest one to form the fibrils, while WT is the slowest? It should be clarified.
- L128-L137, this paragraph normally describes briefly the work did, which are missing in this manuscript.
- L161, proteins are usually dissolved in a pH buffer, while here in pure water. The underlying rational should be address. Also condition of the monomer formation should be described. How the authors differentiate monomer and oligomer? They might be in dynamic equilibrium.
- L174, 3.5+/- 0.6 x 102 ?
- L198, DMD? Abbreviations should be explained at its first sight in the text.
- L241, HeR?
- L264, Table 2, what is the physical meaning of HeR? How does HeR correlate with SAS? Why oligomer has the lowest HeR value?
- L353, Figure 1?
- L478, literature or further proof of monomer/oligomer formation at the given concentration should be included
- L578, Figure A1.The unit of X-axis of the inset figure is missing.
Author Response
Point 1: L48, is there any other group studied the wide-line 1H NMR?
Response 1: We do not know about any group studying especially wide-line NMR. However, there have been studies on the water content in various materials earlier.
See e.g. R.F. Bartholomew band J.W.H. Schreurs. Wide-line NMR study of protons in hydrosilicate glasses of different water content. (1980). J. Non-Cryst. Solids, 38 & 39, 679-684. DOI: 10.1016/0022-3093(80)90515-3; F. Cornejo and P. Chinachoti. NMR in foods: opportunity and challenges. in: Magnetic resonance in foods. (eds: P.S. Belton, A.M. Gil, G.A. Webb, D.Rutledge), The Royal Society of Chemistry, Cambridge, UK, 2003; I.J. van den Dries, D. van Dusschoten, M.A. Hemminga. (1998). Mobility in Maltose-Water Glasses Studied with 1H NMR. J. Phys. Chem. B, 102, 10483-10489. DOI: 10.1021/jp982718v; K. Izutsu, S. Yoshioka, S. Kojima. (1995). Effect of cryoprotectants on the eutectic crystallization of NaCl in frozen solutions studied by differential scanning calorimetry (DSC) and broad-line pulsed NMR. Chem. Pharm. Bull., 43, 1804-1806. DOI: 10.1248/cpb.43.1804; C. Garvey, I. Parker, S. George, A. Whittaker. (2006). The hydration of paper studied with solid-state magnetisation-exchange H-1 NMR spectroscopy. Holzforschung, 60, 409-416. DOI: 10.1515/HF.2006.064; J. Clements, G.R. Davies, I.M. Ward. A broad line n.rn.r. study of oriented poly(vinylidene fluoride). (1985). Polymer, 26, 208-218. DOI: 10.1016/0032-3861(85)90032-1.
Point 2: L78, alpha-sheet structures?
Response 2: Yes, see publications J.A. Fauerbach, D.A. Yushchenko, S.H. Shahmoradian, W. Chiu, T.M. Jovin, E.A. Jares-Erijman. (2012). Supramolecular non-amyloid intermediates in the early stages of a-synuclein aggregation. Biophys. J., 102, 1127-1136. DOI: 10.1016/j.bpj.2012.01.051; A. Balupuri, K. Choi, N.S. Kang. (2019). Computational insights into the role of α-strand/sheet in aggregation of α-synuclein. Sci. Rep., 9, 59. DOI: 10.1038/s41598-018-37276-1
Point 3: L81-82, did the authors mean that among all the WT and mutants, A53T mutant is the fastest one to form the fibrils, while WT is the slowest? It should be clarified.
Response 3: Yes. The sentence should be read as “A53T αS forms fibrils the fastest in vitro [21], and WT αS is the slowest in fibril formation of all the WT and familial mutant variants (A30P, E46K, A53T).”
Point 4: L128-L137, this paragraph normally describes briefly the work did, which are missing in this manuscript.
Response 4: The texts concerning the dynamic parameters of the melting diagram are organized into a new chapter entitled “2.4. Dynamic parameters and comparison” at the end of section 2. Results.
Point 5: L161, proteins are usually dissolved in a pH buffer, while here in pure water. The underlying rational should be address. Also condition of the monomer formation should be described. How the authors differentiate monomer and oligomer? They might be in dynamic equilibrium.
Response 5: The protein was dissolved in water to prevent the disturbing effects any other substance would cause. Monomers were separated from oligomers before the measurement by filtering the solution through a 100 kDa membrane. Fibril formation was induced in TRIS buffer and followed with ThT fluorescence. Oligomers were separated at the lag phase of fibril growth and the formation of the fibrils was also checked by electron microscopy. To prevent the disturbances of the additives, samples were dialyzed into pure water before NMR measurements. A detailed explanation of the methods is added to the manuscript.
Point 6: L174, 3.5+/- 0.6 x 102 ?
Response 6: It is correct.
Point 7: L198, DMD? Abbreviations should be explained at its first sight in the text.
Response 7: DMD = differential form of the melting diagram, it is inserted in the text
Point 8: L241, HeR?
Response 8: HeR = heterogeneity ratio, it is inserted in the text
Point 9: L264, Table 2, what is the physical meaning of HeR? How does HeR correlate with SAS? Why oligomer has the lowest HeR value?
Response 9: See our earlier publication K. Tompa, M. Bokor, P. Tompa. (2018). The melting diagram of protein solutions and its thermodynamic interpretation. Int. J. Mol. Sci., 19, 3571. DOI: 10.3390/ijms19113571
HeR = (1 – Tfne)/(1 – Tfno), in which (1 – Tfne) and (1 – Tfno) give the measured distances from the melting point of ice. HeR gives order parameter type specification for what extent of the surface of the protein molecule (SAS) can be regarded as showing heterogeneous potential energy distribution (disordered) in terms of water binding.
The oligomers have the lowest HeR value because their solvent accessible surface area is most homogenous in potential barriers governing the motions of hydration water, i.e. in the energy of protein-water interactions. (1 – Tfno) is large compared to (1 – Tfne), while this relation is reversed in the monomers.
Point 10: L353, Figure 1?
Response 10: We have unified the figures into one multi-panel figure. That way, the problem is solved.
Point 11: L478, literature or further proof of monomer/oligomer formation at the given concentration should be included
Response 11: Literature: Uversky, V. N., Li, J. and Fink, A.L. (2001) Evidence for a Partially Folded Intermediate in α-Synuclein Fibril Formation J. Biol. Chem. 276: 10737-10744. doi: 10.1074/jbc.M010907200
Point 12: L578, Figure A1.The unit of X-axis of the inset figure is missing.
Response 12: It is the same as in the main figure, time in ms, corrected.
Reviewer 2 Report
Review of IJMS manuscript 777044
Summary & background:
Bokar et al describe the reanalysis of wide field 1H NMR data obtained for wildtype and mutant forms of a-synuclein (aS) using a formalism developed in a previous IJMS paper to describe this type of data. Tompa and coworkers have pioneered a technique for assessing the amount of mobile water interacting with partially frozen protein solutions using wide field 1H NMR. The technique relies on the ability of NMR to quantify mobile water in a sample and compare it to the total water present, therefore deriving the amount (or percent) of hydration water. By coupling this with controlled melting of the sample they can investigate the portion of water which becomes mobile as a function of the available thermal activation energy thus deriving information about the binding energy of different protein-bound waters. The diagram plotting the amount of mobile water as a function of temperature below the melting point is referred to as a melting diagram or MD, and the derivative of this is the derivative melting diagram or DMD.
In a previous paper (Hazy et al. 2011 - reference #1) the same group analysed the MDs of the intrinsically disordered protein α-synuclein, implicated in the prion-like Parkinson’s disease, and a familial mutant of aS A53T which increases likelihood of PD and shows more rapid aggregation in the fibrils, a characteristic process in disease progression. Hazy et al 2011 demonstrated that unlike globular proteins which show an MD with low temperature of initial melting (~ -50C), and a long plateau followed by a steep rise just below the melting temperature of bulk water, both WT and A53T aS show a higher initial melting temperature (~ -40C), similar to another previously studied IDP. Furthermore, after this initial melting they showed a very small plateau region in the MD followed by a region with steadily increasing amount of mobile water, which was ascribed to the presence of a heterogenous protein-water interface due to the highly exposed protein surface of the IDP. Interestingly they saw a difference between WT and variant A53T aS with A53T showing a lower onset of melting, and a consistently higher level of hydration. They interpreted this difference to be caused by the more open structure of A53T, which was indicated also by results reported in the literature based on other techniques, and implicated in the mutants more rapid aggregation.
Two important facts should be mentioned with regard to the current manuscript: Hazy et al performed their experiments at two concentrations (25mg/ml) where they obtained the results I have just summarized, and at 50mg/ml. At the higher concentration they observed an MD, with low onset temperature of mobile water, a lower hydration level, and much slower rise in amount of hydration water with temperature- behavior much more similar to a globular protein. Hazy et al. also perform experiments with amyloids at 25mg/ml, where they also see a low onset temperature and slower rise in amount of mobile water with temperature. They also observed no difference between the MD of WT or mutant aS amyloid. They suggested that the reason for the similarity between amyloid and globular protein MDs was because amyloids have significant amounts of structure like a globular protein. They explained the similarity between the MD of amyloid aS and the 50mg/ml aS sample might be caused by the formation of amyloid in the concentrated aS sample before or during the measurement.
General assessment:
The major concerns I have regarding the submitted manuscript are (a) it does not make sufficiently clear what is new information relative to the Hazy et al paper, and (b) there are sections of the paper which are very difficult to follow because of unclear wording, repeated content and a lack of structuring of the content. I think a rewriting of the paper to simultaneously address these points would result on a more succinct (shorter paper) which was more understandable, and more useful to the IJMS readership. (c) Finally, I had a concern about the agreement between the previous work and the current manuscript. Namely, in the submitted manuscript the authors worked at a concentration of 50mg/ml and yet in the Hazy et al paper this concentration resulted in very different looking data. Figure 2 of Hazy et al shows an onset of mobile water below -50C, and yet in the submitted manuscript monomers show onset of mobile water at -40C (or even -35C for WT). The data in Figure 2 of Hazy et al. seem to match better with Figure 2 of the submitted manuscript suggesting that in that Hazy et al paper they were actually measuring oligomers in the 50mg/ml aS samples. Furthermore the hydration level of amyloids in Figure 3 line 352 of the submitted manuscript (incorrectly labeled Figure 1) seems to be much lower than the hydration level reported at the same temperatures in Hazy et al.
If I have appraised correctly, the distinguishing features of the new manuscript are:
1) it systematically compares amyloid, oligomer, and monomer state of WT and A53T aS, and
2) it applies the formalism for analysing MD and DMDs introduced in Tompa K et al. 2018, IJMS (reference #3)
To authors:
Therefore to improve the manuscript you should emphasize these two aspects. With respect to point 1 above, I think you should point out that you have revisited aS using improved protocols for preparing aS in these three key states, and you have applied a more quantitative analysis method (Tompa K etal 2018) which allow you to make direct comparison of these states for both WT and A54T aS. The preparation of samples and characterization of their states should receive more attention in the methods to make clear how this is an improvement over your previous work - currently there is only very cursory information given on how protein was prepared in these three states and how the conditions were analysed to affirm that the protein is indeed in these 3 different states. This fact should receive brief mention in the abstract and also be discussed in the introduction.
In the following I go into some more detail on how improvements could be made:
Try to rewrite the abstract to be more readable and engaging. Currently it reads like a list of results. Ofcourse it should be compact, but the reader should understand what it is the paper reports that is important or distinguishes it.
It should be clearly stated in the introduction what has been done previously and what the goal of this paper is, i.e. what will be added to the picture relative to the previous publication. The Results should be shortened to only describe results without discussing them, and the discussion should then explain how the new analysis method has contributed to a better understanding of the questions raised. The discussion requires work with someone with good English skills. Much of it is not written grammatically or his very difficult to understand.
The paper was very hard to read because of the order of exposition. It begins by reviewing previous work, but has not introduced the protein or why it is of interest to study.
The first paragraph does not set up a background for this study. What is the interest in α-synuclein? What open questions are important that you want to answer? Why use your technique? What new analysis methods are you using to expand on your previous work and why is this important? What is the relevance of the A53T mutant?
The protein is discussed too late (in paragraph starting at line 63).
I propose that you start with this paragraph, which establishes the importance of α-synuclein as a subject of study using some text from the first paragraph. E.g.
“The protein α-synuclein (αS) plays an essential role in in Parkinson’s disease (PD)… Human αS is a 140-residue, highly conserved presynaptic protein that is abundant in various regions of the brain [11,12]. Structurally, purified αS is a monomeric intrinsically disordered protein (IDP) at neutral pH [1,13].” “IDP refers to proteins without fixed three-dimensional structures under physiological conditions.” “Due to their specific amino acid composition, IDPs, in general, have a unique combination of low overall hydrophobicity and large net charge [14]. Though αS is an IDP, it shows a limited amount of secondary structure [15-17] and it is slightly more compact than a random coil [16,18]. Furthermore high-resolution NMR analysis revealed that it exhibits a region with a preference for helical conformation [18].”
At this point you should introduce the central question you want to answer with your study. For example by describing the phenomenon of aggregation and the role water could play in this. Then pointing out that wide line NMR can furnish information about the energetics of the hydration shell. Describe briefly the results of your previous work and the questions, which were left unanswered by that work, and how this work will expand on the previous work. This would contain information from paragraphs 1 and 2 of your manuscript.
You discuss data before presenting it.
The oligomer and amyloid data should have been presented before you compare the results obtained with these conditions and those of the monomer. I propose that you show the MD and DMD data of all three states in the same figure. This will allow a side-by-side comparison, and a consolidation of the discussion and comparison of the three states (amyloid, oligomer, monomer).
The figures can be improved as follows:
Show all data (amyloid, oligomer, monomer) with a single horizontal scale at the bottom (temp & Tfn) and top (Ea). Increase size of plots to make them larger since there is room. Reduce the size of data points and use non-filled symbols for one of the two datasets (the one on top) so that all datapoints of both datasets are visible. See Hazy et al. 2011, or Tompa K et al. 2010 for good examples. Do not fill under the curve for DMD plots and use a curve for both data so they are easy to distinguish. Mark the position of Ea,0 which is referred to in the text on the plots. Figure 2. Correct color of green arrow to be same as color as datapoints. Figure 3 should have its caption title changed to Figure 3 (not Figure 1), if the figures remain separate. But If you consolidate figures it would be part c. (DMDS would be e,f,g).
Table 1.
Generally this table is difficult to read because of lack of structuring introduce some space between the different Ea,n data rows. Define the terms which are shown here (or refer to the methods, appendix).
Table 2.
Structure this by introducing space between data for monomer, oligomer, amyloid, and between the WT and A53T. amyloid of WT and A53T were measured as separate datasets and their results should be reported in separate rows not lumped together.
Minor text:
Proposed insertions are bold, proposed deletions are struck out. Line numbers are indicated.
78: α-sheet structures
I assume that β-sheet structures is meant?
82: αS oligomers, which are thought to be the precursors of fibrils is are more toxic than mature fibrils
89: exposed hydrophobicity hydrophobic surface
95: simulations suggest a hydrate hydration shell
96: The differences between WT and A53T
What differences? The differenes in exposed hydrophobic surface? In hydration? Please be more specific here.
101: water binding heterogeneity
Please describe what this means somewhere preferably at the point where you introduce the technique and its unique capability in regard to the questions you want to answer.
The following text occurs out of context. It is not the place to discuss these considerations. This should be discussed at the place you introduce the method (see above).
101: The NMR
102 measurements presented in reference [1], were not focused on the structures of the
103 proteins. They presented an interpretation process based on the measurement of the
104 mobility of hydration water. The interpretation of the resulting melting diagrams
105 (MD, amount of mobile hydration vs. normalized fundamental temperature)
106 informs about the thermodynamic nature of the SAS of the protein.
123: The dynamic order parameter of the MD, i.e. the heterogeneity ratio
What is “MD” here? Are you referring to molecular dynamics? Or is this the abbreviation of another term?
143: The phenomenon of motional
143 narrowing in NMR therefore, is a good indicator of mobile water.
Motional narrowing of what? It is unclear what signals motional narrowing is meant in this statement. For example clearer would be “The phenomenon of motional narrowing of the wideline 1H NMR signal of the water therefore, is a good indicator of mobile water. Also under what conditions would this be measured. Currently the entire description of MD diagrams is relegated to methods. But a brief description of what is being measured and how would it is analysed or interpreted would be helpful at the start of results.
151 The extent of
152 the exposed hydrophobicity
Hydrophobicity is a property (how hydrophobic is an amino acid?) but here you appear to be discussing the extent (or area) of the exposed hydrophobic surface. Please use an appropriate term to describe the extensible property.
156 If the amino acid composition of a protein is known,
157 predictions can be made of the amount of bound water [30,32].
Generally true? Or is this true only for a disordered or unfolded protein?
163 a relatively high potential barrier
164 value as the lowest (Ea,0, Table 1).
As the lowest what? The object that is being discussed is missing from this sentence.
Surely you mean per amino acid? “per amino acid sequence” does not make any sense.
168 The MDs of WT and A53T αS monomers show rapidly changing trends as a
169 function of potential barriers governing the motion of hydration water, except for a
170 short constant region at the start of mobility (Fig. 1).
It is not clear what is meant by “as a function of potential barriers” since the MDs are plotted as a function of temperature. And what is meant by “the start of mobility” It would be helpful to briefly introduce the diagrams and describe what it is that is displayed and why the diagram looks as it does.
170 short constant region at the start of mobility (Fig. 1). The amount of mobile water
171 molecules, h = 0.44(8) is enough to have functional proteins with approximately
172 monolayer coverage at the smallest potential barriers below Ea,1, where constant h
173 region can be found.
The parameter h is introduced and used before explaining what it is. It does not appear in the diagram.
178 Earlier NMR measurements and
179 molecular dynamics simulations for individual amino acid’s hydration yield 4(1)·102
180 water molecules per amino acid sequence of the whole protein [32-34].
183 The
184 amount of homogeneously bound hydration water, n(Tfn0) = n(Ea,0) is as high as if
185 every possible hydrogen bond site were occupied by a water molecule. This refers
186 to open structure.
Where is this value on the MD plot? It should be indicated as it is referred to also later in the text.
191 Water in interaction with A53T αS gets in motion at a lower potential barrier
192 compared to the WT variant (Fig. 1). This fact indicates that A53T αS interact with
193 water weaker more weakly, with looser protein-water bonds are looser.
195 To support this statement, see the
196 MDs of the two globular proteins, UBQ and BSA dissolved in water and the MDs of
197 IDPs [2,3]. The occurrence of secondary structures in IDPs is not unexpected [35].
It would be better to actually say what is observed for UBQ and BSA vs. for IDPs. For example describing a long plateau region for the two globular proteins which correlates with the amount of secondary structure in the global proteins, vs. IDPs with no secondary structure where this feature is not observed in the MD. It is then also confusing to say “the occurrence of secondary structures in IDPs is not unexpected” immediately after using IDPs to demonstrate the absence of this feature. So there should be some transition: e.g. “Never-the-less for some IDPs including aS, secondary structure has been reported using other techniques, so the results we obtained for aS are not entirely unexpected.”
198 These regions appear as spikes and subsequent zero values in DMDs. Consequently,
199 only one type of potential barrier or protein-water interaction is responsible for the
200 motion of water molecules in this energy region.
Please define DMD before using it.
200 The type of the interaction is not
201 van der Waals here, because it lacks continuous distance dependence; this behaviour
202 suggests the formation of hydrogen bonds.
Better:
The narrow energy spectrum of this feature is not consistent with a van der Waals interaction, which would have a wider range of interaction energies, and suggests instead the formation of hydrogen bonds.
203 The plateau region (constant value of h) is wider for A53T αS but the Ea,1 values
204 are the same for both variants (Table 1).
This confused me. Up to now I thought the plateau region referred to the constant n value between Ea,1 and Ea,2 on the plot, but since the separation of these two points is indistinguishable for WT and A53T, I conclude that what is meant is the gap between the first spike and Ea,1. Then it would be best to include Ea,0 in the plot and refer to it in the text.
Please define all the parameters described in Table 1.
216 Elevation of MD for αS monomers in the linear section (Ea,1 < Ea < Ea,2) is more than
217 three times steeper
Elevation is not the word you want here. Could it be you are talking about the slope?
Where is the MD data for amyloid and for aggregates? I see only a figure showing MD data for monomers. This is why I propose to consolidate the data into a single figure and discuss them together.
217 The
218 introduction appearance of a heterogeneous distribution of potential barriers at Ea,1 is connected
219 to with a general change in the motional state of the hydration water.
219 As a blanket result, In general it
220 was found for both globular proteins and IDPs that the dynamic contributions to the
221 quasi-elastic neutron scattering spectra [29] all change their thermal trend around
222 Ea,1. At the higher hydration levels, not only hydrogen bonds are responsible for
223 protein-water interactions but van der Waals forces also are acting. Above Ea,2 A section of
224 quadratic trend was detected in the MDs for both αS monomer variants above Ea,2 showed a quadratic trend
225 (Fig. 1).
229 The αS monomers have the highest ratio of exposed hydrophilic groups with five
230 times more moving water in their first hydration shell than the αS amyloids.
Where should I be seeing the data supporting this statement? Is it in Table 1? Which parameter? Please refer to the data discussed.
The αS monomer-water interaction energy is most diversified shows a wider distribution than either the amyloids or the aggregates.
The data for the amyloids and aggregates need to be presented already if such a comparison is made.
235 The difference
236 between the MD of the WT and A53T αS monomers is very minute (Fig. 1), though
237 the A53T mutant is steadily more heterogeneous with higher hydration levels.
What is meant by “more heterogeneous”? What should I be looking at to see this difference – either describe the feature in the figure or a parameter on the table or both.
238 WT and A53T αS monomers reach the far highest level of hydration compared
239 to oligomers and amyloids at the melting point of bulk water, h(Ea = 6.01 kJ mol-1,
240 monomer) = 3.31(7) on the average, substantiating the most open structure for them.
“far highest level … compared to” is not understandable. Do you mean h is largest at the melting point of water? Possibly the way to state this is “Both WT and A53T αS monomers reach their highest hydration level at the melting point of bulk water, h... exceeding the values observed for oligomers and amyloids, and indicating a mostly solvent exposed polypeptide chain.”
The discussion requires work with someone with good English skills. Much of it is not written grammatically or his very difficult to understand. It also does not present the results in the context of previous work or indicate clearly what is new knowledge. To me the abstract and discussion have similar problems. They are a list of facts but do not put things together in a clearly discernable line of argument.
Author Response
Please see the attachment. (Modified manuscript.)
Point 1: Therefore, to improve the manuscript you should emphasize these two aspects. With respect to point 1 above, I think you should point out that you have revisited aS using improved protocols for preparing aS in these three key states, and you have applied a more quantitative analysis method (Tompa K etal 2018) which allow you to make direct comparison of these states for both WT and A54T aS. The preparation of samples and characterization of their states should receive more attention in the methods to make clear how this is an improvement over your previous work - currently there is only very cursory information given on how protein was prepared in these three states and how the conditions were analyzed to affirm that the protein is indeed in these 3 different states. This fact should receive brief mention in the abstract and also be discussed in the introduction.
Response 1: A more detailed explanation of the sample preparation is added to the text. In short: monomers were measured directly after dissolving the protein in ultrapure water and filtering to remove already formed oligomers, while amyloid formation was induced in a TRIS buffer with CuCl2 at 37 oC. Amyloid formation was followed by ThT fluorescence and oligomers were separated in the lag phase of fibril growth. Oligomer characterization was done by the NMR measurements and comparing the results of known oligomer characteristics. No seeding was applied, and amyloid fibrils were checked with electron microscopy.
Point 2: Try to rewrite the abstract to be more readable and engaging. Currently it reads like a list of results. Of course it should be compact, but the reader should understand what it is the paper reports that is important or distinguishes it.
Response 2: The abstract is rewritten.
Point 3: It should be clearly stated in the introduction what has been done previously and what the goal of this paper is, i.e. what will be added to the picture relative to the previous publication. The Results should be shortened to only describe results without discussing them, and the discussion should then explain how the new analysis method has contributed to a better understanding of the questions raised. The discussion requires work with someone with good English skills. Much of it is not written grammatically or his very difficult to understand.
Response 3: The introduction is rearranged.
Point 4: The paper was very hard to read because of the order of exposition. It begins by reviewing previous work, but has not introduced the protein or why it is of interest to study.
The first paragraph does not set up a background for this study. What is the interest in α-synuclein? What open questions are important that you want to answer? Why use your technique? What new analysis methods are you using to expand on your previous work and why is this important? What is the relevance of the A53T mutant?
The protein is discussed too late (in paragraph starting at line 63).
I propose that you start with this paragraph, which establishes the importance of α-synuclein as a subject of study using some text from the first paragraph. E.g.
“The protein α-synuclein (αS) plays an essential role in Parkinson’s disease (PD)… Human αS is a 140-residue, highly conserved presynaptic protein that is abundant in various regions of the brain [11,12]. Structurally, purified αS is a monomeric intrinsically disordered protein (IDP) at neutral pH [1,13].” “IDP refers to proteins without fixed three-dimensional structures under physiological conditions.” “Due to their specific amino acid composition, IDPs, in general, have a unique combination of low overall hydrophobicity and large net charge [14]. Though αS is an IDP, it shows a limited amount of secondary structure [15-17] and it is slightly more compact than a random coil [16,18]. Furthermore high-resolution NMR analysis revealed that it exhibits a region with a preference for helical conformation [18].”
At this point you should introduce the central question you want to answer with your study. For example by describing the phenomenon of aggregation and the role water could play in this. Then pointing out that wide line NMR can furnish information about the energetics of the hydration shell. Describe briefly the results of your previous work and the questions, which were left unanswered by that work, and how this work will expand on the previous work. This would contain information from paragraphs 1 and 2 of your manuscript.
You discuss data before presenting it.
The oligomer and amyloid data should have been presented before you compare the results obtained with these conditions and those of the monomer. I propose that you show the MD and DMD data of all three states in the same figure. This will allow a side-by-side comparison, and a consolidation of the discussion and comparison of the three states (amyloid, oligomer, monomer).
Response 4: The manuscript has been modified accordingly. (See attachment.)
Point 5: The figures can be improved as follows:
Show all data (amyloid, oligomer, monomer) with a single horizontal scale at the bottom (temp & Tfn) and top (Ea). Increase size of plots to make them larger since there is room. Reduce the size of data points and use non-filled symbols for one of the two datasets (the one on top) so that all datapoints of both datasets are visible. See Hazy et al. 2011, or Tompa K et al. 2010 for good examples. Do not fill under the curve for DMD plots and use a curve for both data so they are easy to distinguish. Mark the position of Ea,0 which is referred to in the text on the plots. Figure 2. Correct color of green arrow to be same as color as datapoints. Figure 3 should have its caption title changed to Figure 3 (not Figure 1), if the figures remain separate. But If you consolidate figures it would be part c. (DMDS would be e,f,g).
Response 5: The figures are corrected – one combined figure.
Point 6: Table 1.
Generally this table is difficult to read because of lack of structuring introduce some space between the different Ea,n data rows. Define the terms which are shown here (or refer to the methods, appendix).
Response 6: Table 1. has been structured. A reference to Appendix A was added.
Point 7: Table 2.
Structure this by introducing space between data for monomer, oligomer, amyloid, and between the WT and A53T. amyloid of WT and A53T were measured as separate datasets and their results should be reported in separate rows not lumped together.
Response 7: Table 2. has been structured.
Proposed insertions are bold, proposed deletions are struck out. Line numbers are indicated.
Point 8: 78: α-sheet structures
I assume that β-sheet structures is meant?
Response 8: (line 86 after reorganizing the text) No, there are also a-sheets. See publications J.A. Fauerbach, D.A. Yushchenko, S.H. Shahmoradian, W Chiu, T.M. Jovin, E.A. Jares-Erijman. (2012). Supramolecular non-amyloid intermediates in the early stages of a-synuclein aggregation. Biophys. J., 102, 1127-1136. DOI: 10.1016/j.bpj.2012.01.051; A. Balupuri, K. Choi, N.S. Kang. (2019). Computational insights into the role of α-strand/sheet in aggregation of α-synuclein. Sci. Rep., 9, 59. DOI: 10.1038/s41598-018-37276-1
Point 9: 82 αS oligomers, which are thought to be the precursors of fibrils is are more toxic than mature fibrils
Response 9: Accepted. (line 89 after reorganizing the text)
Point 10: 89: exposed hydrophobicity hydrophobic surface
Response 10: Accepted. (line 95 after reorganizing the text)
Point 11: 95: simulations suggest a hydrate hydration shell
Response 11: Accepted. (line 104 after reorganizing the text)
Point 12: 96: The differences between WT and A53T
What differences? The differences in exposed hydrophobic surface? In hydration? Please be more specific here.
Response 12: The sentence has been rewritten. (lines 119-121 after reorganizing the text)
Point 13: 101: water binding heterogeneity
Response 13: Accepted. (line 103-104 after reorganizing the text)
Point 14: Please describe what this means somewhere preferably at the point where you introduce the technique and its unique capability in regard to the questions you want to answer.
Response 14: An explanatory sentence is inserted to lines 537-539.
Point 15: The following text occurs out of context. It is not the place to discuss these considerations. This should be discussed at the place you introduce the method (see above).
101: The NMR
102 measurements presented in reference [1], were not focused on the structures of the
103 proteins. They presented an interpretation process based on the measurement of the
104 mobility of hydration water. The interpretation of the resulting melting diagrams
105 (MD, amount of mobile hydration vs. normalized fundamental temperature)
106 informs about the thermodynamic nature of the SAS of the protein.
Response 15: The text was modified and transposed.
Point 16: 123: The dynamic order parameter of the MD, i.e. the heterogeneity ratio
What is “MD” here? Are you referring to molecular dynamics? Or is this the abbreviation of another term?
Response 16: It means melting diagram, it is explained in lines 55-56 (new numbering) or 105 in the original text.
Point 17: 143: The phenomenon of motional
143 narrowing in NMR therefore, is a good indicator of mobile water.
Motional narrowing of what? It is unclear what signals motional narrowing is meant in this statement. For example clearer would be “The phenomenon of motional narrowing of the wide-line 1H NMR signal of the water therefore, is a good indicator of mobile water. Also under what conditions would this be measured. Currently the entire description of MD diagrams is relegated to methods. But a brief description of what is being measured and how would it is analyzed or interpreted would be helpful at the start of results.
Response 17: “Motional narrowing” is a widely used, commonplace idea in NMR. In physics and chemistry, motional narrowing is a phenomenon where a certain resonant frequency has a smaller linewidth than might be expected, due to motion in an inhomogeneous system, e.g. liquids. The phenomenon of motional narrowing corresponds to a sort of averaging out of the effect of the interaction with the static magnetic field when the fluctuations of the magnetic field become sufficiently rapid [C.P. Slichter. Principles of magnetic resonance. in Springer Series in Solid-State Sciences 1, ed. P. Fulde, Springer-Verlag, Berlin Heidelberg New York London Tokyo Hong Kong, 1983]. Any motion must result in the motional narrowing of the spectral lines.
Point 18: 151 The extent of
152 the exposed hydrophobicity
Hydrophobicity is a property (how hydrophobic is an amino acid?) but here you appear to be discussing the extent (or area) of the exposed hydrophobic surface. Please use an appropriate term to describe the extensible property.
Response 18: It is modified to “The extent of hydrophobic surface…”.
Point 19: 156 If the amino acid composition of a protein is known,
157 predictions can be made of the amount of bound water [30,32].
Generally true? Or is this true only for a disordered or unfolded protein?
This is generally true to any protein.
163 a relatively high potential barrier
164 value as the lowest (Ea,0, Table 1).
As the lowest what? The object that is being discussed is missing from this sentence.
Surely you mean per amino acid? “per amino acid sequence” does not make any sense.
Response 19: This sentence is modified as “at a relatively high potential barrier value as the lowest potential barrier value where mobile hydration water exists (Ea,0, Table 1)”
Point 20: 168 The MDs of WT and A53T αS monomers show rapidly changing trends as a
169 function of potential barriers governing the motion of hydration water, except for a
170 short constant region at the start of mobility (Fig. 1).
It is not clear what is meant by “as a function of potential barriers” since the MDs are plotted as a function of temperature. And what is meant by “the start of mobility” It would be helpful to briefly introduce the diagrams and describe what it is that is displayed and why the diagram looks as it does.
Response 20: Potential barriers are equivalent to temperatures as they are (absolute) temperatures normalized with the melting point of ice and scaled with the melting heat of ice, Ea = T/273.15 K·6.01 kJ/mol.
See reference [15], there every idea is thoroughly introduced and explained. We haven’t wanted repetitions.
Point 21: 170 short constant region at the start of mobility (Fig. 1). The amount of mobile water
171 molecules, h = 0.44(8) is enough to have functional proteins with approximately
172 monolayer coverage at the smallest potential barriers below Ea,1, where constant h
173 region can be found.
The parameter h is introduced and used before explaining what it is. It does not appear in the diagram.
Response 21: The sentence is modified as “The amount of mobile water molecules, h = 0.44(8) is enough to have functional proteins with approximately monolayer coverage at the smallest potential barriers below Ea,1, where constant h region can be found (for definition of hydration see section 4.3. Interpretation methods of the melting diagram).” Hydration appears on the diagram on right side ordinates.
Point 22: 178 Earlier NMR measurements and
179 molecular dynamics simulations for individual amino acid’s hydration yield 4(1)·102
180 water molecules per amino acid sequence of the whole protein [32-34].
183 The
184 amount of homogeneously bound hydration water, n(Tfn0) = n(Ea,0) is as high as if
185 every possible hydrogen bond site were occupied by a water molecule. This refers
186 to open structure.
Where is this value on the MD plot? It should be indicated as it is referred to also later in the text.
Response 22: The predicted value is marked in Fig. 1. There has been put a reference to it in the text, lines 183-184.
Point 23: 191 Water in interaction with A53T αS gets in motion at a lower potential barrier
192 compared to the WT variant (Fig. 1). This fact indicates that A53T αS interact with
193 water weaker more weakly, with looser protein-water bonds are looser.
Response 23: The sentence is mended as “This fact indicates that A53T αS interact with water more weakly, its protein-water bonds are looser.”. New line numbers are 198-199.
Point 24: 195 To support this statement, see the
196 MDs of the two globular proteins, UBQ and BSA dissolved in water and the MDs of
197 IDPs [2,3]. The occurrence of secondary structures in IDPs is not unexpected [35].
It would be better to actually say what is observed for UBQ and BSA vs. for IDPs. For example describing a long plateau region for the two globular proteins which correlates with the amount of secondary structure in the global proteins, vs. IDPs with no secondary structure where this feature is not observed in the MD. It is then also confusing to say “the occurrence of secondary structures in IDPs is not unexpected” immediately after using IDPs to demonstrate the absence of this feature. So there should be some transition: e.g. “Never-the-less for some IDPs including aS, secondary structure has been reported using other techniques, so the results we obtained for aS are not entirely unexpected.”
Response 24: The following sentence has been inserted between the quoted sentences. “The globular proteins have MDs consisting of almost only temperature-independent hydration while the IDPs have constantly increasing MDs with no or very narrow temperature-independent hydration.”
Point 25: 198 These regions appear as spikes and subsequent zero values in DMDs. Consequently,
199 only one type of potential barrier or protein-water interaction is responsible for the
200 motion of water molecules in this energy region.
Please define DMD before using it.
Response 25: It is defined as “These regions appear as spikes and subsequent zero values in differential forms of melting diagrams (DMDs).”.
Point 26: 200 The type of the interaction is not
201 van der Waals here, because it lacks continuous distance dependence; this behaviour
202 suggests the formation of hydrogen bonds.
Better:
The narrow energy spectrum of this feature is not consistent with a van der Waals interaction, which would have a wider range of interaction energies, and suggests instead the formation of hydrogen bonds.
Response 26: Accepted.
Point 27: 203 The plateau region (constant value of h) is wider for A53T αS but the Ea,1 values
204 are the same for both variants (Table 1).
This confused me. Up to now I thought the plateau region referred to the constant n value between Ea,1 and Ea,2 on the plot, but since the separation of these two points is indistinguishable for WT and A53T, I conclude that what is meant is the gap between the first spike and Ea,1. Then it would be best to include Ea,0 in the plot and refer to it in the text.
Response 27: The Ea,i points are indicated on the graphs and it was explicitly given in the text. “(constant value of h, Ea,0 ≤ Ea ≤ Ea,1)”
Point 28: Please define all the parameters described in Table 1.
Response 28: The sentence “For the detailed definitions of the parameters, see Supplemental Information of [8].” is inserted in the legend of Table 1.
Point 29: 216 Elevation of MD for αS monomers in the linear section (Ea,1 < Ea < Ea,2) is more than
217 three times steeper
Elevation is not the word you want here. Could it be you are talking about the slope?
Response 29: It is changed to “slope”.
Point 30: Where is the MD data for amyloid and for aggregates? I see only a figure showing MD data for monomers. This is why I propose to consolidate the data into a single figure and discuss them together.
Response 30: Data are consolidated into a single, multi-panel figure.
Point 31: 217 The
218 introduction appearance of a heterogeneous distribution of potential barriers at Ea,1 is connected
219 to with a general change in the motional state of the hydration water.
219 As a blanket result, In general it
220 was found for both globular proteins and IDPs that the dynamic contributions to the
221 quasi-elastic neutron scattering spectra [29] all change their thermal trend around
222 Ea,1. At the higher hydration levels, not only hydrogen bonds are responsible for
223 protein-water interactions but van der Waals forces also are acting. Above Ea,2 A section of
224 quadratic trend was detected in the MDs for both αS monomer variants above Ea,2 showed a quadratic trend
225 (Fig. 1).
Response 31: Accepted.
Point 32: 229 The αS monomers have the highest ratio of exposed hydrophilic groups with five
230 times more moving water in their first hydration shell than the αS amyloids.
Where should I be seeing the data supporting this statement? Is it in Table 1? Which parameter? Please refer to the data discussed.
Response 32: Reference is inserted.
Point 33: The αS monomer-water interaction energy is most diversified shows a wider distribution than either the amyloids or the aggregates.
Response 33: This sentence remains as it is because the proposed statement is not true.
The data for the amyloids and aggregates need to be presented already if such a comparison is made.
These data are presented in Figure 1 as panels (c-f).
Point 34: 235 The difference
236 between the MD of the WT and A53T αS monomers is very minute (Fig. 1), though
237 the A53T mutant is steadily more heterogeneous with higher hydration levels.
What is meant by “more heterogeneous”? What should I be looking at to see this difference – either describe the feature in the figure or a parameter on the table or both.
Response 34: The sentence is modified: “The difference between the MD of the WT and A53T αS monomers is very minute (Fig. 1(a)), though the A53T mutant is steadily more heterogeneous (C, Table 1).”.
Point 35: 238 WT and A53T αS monomers reach the far highest level of hydration compared
239 to oligomers and amyloids at the melting point of bulk water, h(Ea = 6.01 kJ mol-1,
240 monomer) = 3.31(7) on the average, substantiating the most open structure for them.
“far highest level … compared to” is not understandable. Do you mean h is largest at the melting point of water? Possibly the way to state this is “Both WT and A53T αS monomers reach their highest hydration level at the melting point of bulk water, h... exceeding the values observed for oligomers and amyloids, and indicating a mostly solvent exposed polypeptide chain.”
Response 35: Accepted. Modification made at lines 361-364 (reorganized text).

Reviewer 3 Report
Bokor et al. reinterpret in thermodynamic terms previously published data on the hydrate shell of WT and A53T aSyn in monomeric and amyloid forms (Hazy et al. 2011 Biophys J) and extend to oligomeric forms of aSyn. It is my opinion that hydration shells of proteins are largely overlooked, and the work and conclusions presented by the authors are interesting in that respect. I particularly enjoyed the proof that aSyn amyloid fibrils are entropically stabilized compared to disordered monomers, which can be counterintuitive.
My biggest criticism is actually not scientific, but the language is often poor (especially in the introduction), which renders the manuscript difficult to read and makes the argumentation unclear in some cases. I tried to highlight the sentences that were particularly unclear or grammatically incorrect in the minor comments.
A scientific criticism is that all experiment are done in double distilled water, very far from any physiologically relevant medium. The presence of salt, divalent ions as well as molecular crowding is likely to modulate aSyn hydration shell. If possible given the experimental setup, I would advocate for a control experiment in a more relevant buffer and a discussion of the major differences observed (or not) then.
Major comments
Introduction
The introduction needs to be corrected for grammar and language. Some sentences don’t make sense, don’t have a verb. Work needs to be done to account for that.
Lines 33-62 are very poorly written and are mostly repeated afterwards. I would suggest to delete them. This will also allow the introduction to flow better, with first an introduction to aSyn, then to the methods used here. And finally the previous conclusions that were drawn from already published data.
Material and Methods
The authors need to explain how they isolated oligomers and how they formed amyloid fibrils. Did they characterize oligomers? By size, ThT fluorescence? Did they use seeded aggregation to form fibrils? Of which kind of aSyn? Did they use de novo aggregation using shaking, high temperature, low pH, glass beads, organic solvents, etc?
Minor comments
Line 33-36 – It is not clear what is from a previous publication or this present one, please rephrase
Line 34 – previously --> previous
Line 47-52 – sentence reads really odd, please rephrase and ensure readability and grammar
Line 55 – visibly --> visible, delete also
Line 66 – that seems like an overstatement to me, I would advice to lower it down
Line 76 – delete earlier
Line 78 – alpha --> beta
Line 83 – is --> are
Line 91 – I don’t understand what the authors mean by that sentence, please rephrase
Line 102 – delete the comma
Line 109 – delete or rephrase second part of the sentence
Line 115-118 – that sentence is not grammatically correct and does not make sense. I don’t know what the authors mean, it needs to be completely rephrased
Line 123 – comma missing
Line 156-157 – this sentence is not necessary or should come beforehand in the introduction
Line 226-230 – these sentences seem out of place and don’t relate to each other. Please move them to their proper place or delete
Lines 238-263 – it is a bit strange to elaborate on comparing different forms of alphaSyn before presenting results on oligomers and fibrils. Maybe these lines should be moved to the end of the results section? Authors could dedicate a section to a comparison.
Line 272 – it is unclear what the authors mean by “general surface”. Is it geometrical? Chemical properties (hydrophobicity)? Hydration level? Please explain.
Line 327-328 – I am not sure I understand the sentence, please rephrase and clarify
Line 353 – this should be Figure 3
Line 381-382 – I am not sure if that statement makes sense. Do the authors suggest that the termini are more flexible in the fibril than in oligomers? Or is it referring to the fact that oligomers are more globular?
Line 589 – please state what is the “suitable function”
Author Response
Please see the attachment. (Modified manuscript.)
Point 1: Lines 33-62 are very poorly written and are mostly repeated afterwards. I would suggest to delete them. This will also allow the introduction to flow better, with first an introduction to aSyn, then to the methods used here. And finally the previous conclusions that were drawn from already published data.
Response 1: Introduction is rewritten.
Material and Methods
Point 2: The authors need to explain how they isolated oligomers and how they formed amyloid fibrils. Did they characterize oligomers? By size, ThT fluorescence? Did they use seeded aggregation to form fibrils? Of which kind of aSyn? Did they use de novo aggregation using shaking, high temperature, low pH, glass beads, organic solvents, etc?
Response 2: A more detailed explanation of the sample preparation is added to the text. In short: monomers were measured directly after dissolving the protein in ultrapure water and filtering to remove already formed oligomers, while amyloid formation was induced in a TRIS buffer with CuCl2 at 37 oC. Amyloid formation was followed by ThT fluorescence and oligomers were separated in the lag phase of fibril growth. Oligomer characterization was done by the NMR measurements and comparing the results of known oligomer characteristics. No seeding was applied, and amyloid fibrils were checked with electron microscopy.
Minor comments
Point 3: Line 33-36 – It is not clear what is from a previous publication or this present one, please rephrase
Response 3: See the answer to the next comment.
Point 4: Line 34 – previously --> previous
Modify the sentence in lines 34-36, to “We compared the details of the hydration of wild type α-synuclein (WT αS) and its A53T mutant previously by a combination of wide-line 1H NMR, differential scanning calorimetry, and molecular dynamics simulations in publication [1].”
Point 5: Line 47-52 – sentence reads really odd, please rephrase and ensure readability and grammar
Response 4-5: Rephrased and transposed sentences read as “Wide-line 1H NMR results are interpreted in a thermodynamic approach as novelty, in this work. This approach was introduced in our earlier publications [11,12,15-19]. We have made significant advancements in our understanding since publication [4]. We can get information on the energetics of mobile-hydration-water binding properties of proteins due to the advancements. Thermodynamical properties (e.g. fundamental thermal scale, order parameters, surficial potential distribution of proteins) are used in the description.” lines 57-63
Point 6: Line 55 – visibly --> visible, delete also
Response 6: This sentence is rephrased as “Reorienting water molecules bound to a protein molecule also, are mentioned as mobile hydration when they reorient fast enough to be seen as mobile by NMR.” lines 65-67
Point 7: Line 66 – that seems like an overstatement to me, I would advise to lower it down
Response 7: The sentence is modified as “It affects more than 5% of the aged population worldwide (above age 85) and is a challenging problem modern society [1].” lines 36-37
Point 8: Line 76 – delete earlier
Response 8: Accepted, line 82.
Point 9: Line 78 – alpha --> beta
Response 9: It is intently α-sheet and not β-sheet. There are also α-sheets. See publications J.A. Fauerbach, D.A. Yushchenko, S.H. Shahmoradian, W Chiu, T.M. Jovin, E.A. Jares-Erijman. (2012). Supramolecular non-amyloid intermediates in the early stages of α-synuclein aggregation. Biophys. J., 102, 1127-1136. DOI: 10.1016/j.bpj.2012.01.051; A. Balupuri, K. Choi, N.S. Kang. (2019). Computational insights into the role of α-strand/sheet in aggregation of α-synuclein. Sci. Rep., 9, 59. DOI: 10.1038/s41598-018-37276-1
Point 10: Line 83 – is --> are
Response 10: Accepted.
Point 11: Line 91 – I don’t understand what the authors mean by that sentence, please rephrase
Response 11: The sentence is modified to “We have shown earlier that the oligomeric form is ordered, similarly as the globular proteins are [4].” lines 97-98
Point 12: Line 102 – delete the comma
Response 12: The sentence is rephrased. lines 105-108
Point 13: Line 109 – delete or rephrase second part of the sentence
Response 13: This sentence is deleted.
Point 14: Line 115-118 – that sentence is not grammatically correct and does not make sense. I don’t know what the authors mean, it needs to be completely rephrased
Response 14: The sentence should read as follows. “We drew quantitative conclusions about the ratios of the ordered and disordered (more solvent-exposed) surface regions of proteins [15]. The extent of the disordered regions and the energy relations of the protein-water bonds could be determined.”
Point 15: Line 123 – comma missing
Response 15: In British English, “i.e.” is not followed by a comma, so no comma is missing.
Point 16: Line 156-157 – this sentence is not necessary or should come beforehand in the introduction
Response 16: The sentence is deleted.
Point 17: Line 226-230 – these sentences seem out of place and don’t relate to each other. Please move them to their proper place or delete
Response 17: The sentence “Fibril formation is inhibited by intramolecular interactions in the WT αS monomer structure and these interactions are greatly destabilized by the A53T mutation [37], shifting the conformational ensemble of αS to a more open state [38].” is moved to lines 364-367 and is modified as “Intramolecular interactions in the WT αS monomer structure inhibit fibril formation. These interactions are greatly destabilized by the A53T mutation [50], which shifts the conformational ensemble of WT αS monomer to a more open state [51].”.
The sentence “The αS monomers have the highest ratio of exposed hydrophilic groups with five times more moving water in their first hydration shell than the αS amyloids.” is modified as "The αS monomers have the highest ratio of exposed hydrophilic groups with more than five times more moving water in their first hydration shell than the αS amyloids (A = n(Tfno) = n(Tfn1) or h(Ea,o) = h(Ea,1), Table 1.)." and it was transposed to lines 233-236.
Point 18: Lines 238-263 – it is a bit strange to elaborate on comparing different forms of alphaSyn before presenting results on oligomers and fibrils. Maybe these lines should be moved to the end of the results section? Authors could dedicate a section to a comparison.
Response 18: Lines 238-263 are moved to the end of the Results section in a new subsection 2.4. Dynamic parameters and comparison. In addition, the lines 306-308, the sentences in lines 314-316, 378-387 are placed in this section.
Point 19: Line 272 – it is unclear what the authors mean by “general surface”. Is it geometrical? Chemical properties (hydrophobicity)? Hydration level? Please explain.
Response 19: It means chemical properties (hydrophobicity). Wide-line 1H NMR melting diagrams are characteristic of chemical surface properties through the surface’s interactions with hydration water.
Point 20: Line 327-328 – I am not sure I understand the sentence, please rephrase and clarify
Response 20: The sentence is rephrased as “The hydrate shell of the aS amyloids can be mobilized the most easily, which feature is related to the weak bonds holding together the parallel β-sheets.”
Point 21: Line 353 – this should be Figure 3
Response 21: The figures are merged into one multi-panel figure.
Point 22: Line 381-382 – I am not sure if that statement makes sense. Do the authors suggest that the termini are more flexible in the fibril than in oligomers? Or is it referring to the fact that oligomers are more globular?
Response 22: It is referring to the fact that oligomers are more globular. lines 399-401
Point 23: Line 589 – please state what is the “suitable function”
Response 23: A suitable function can be a sum of Gaussian functions or stretched exponentials. lines 603-604

Round 2
Reviewer 1 Report
The authors have addressed my concerns.